# Pediatric Clavicle Fractures and Congenital Pseudarthrosis Unraveled

**DOI:** 10.3390/children9010049

**Published:** 2022-01-03

**Authors:** Lisa van der Water, Arno A. Macken, Denise Eygendaal, Christiaan J. A. van Bergen

**Affiliations:** 1Depeartment of Orthopedic Surgery, Amphia Hospital, 4818 CK Breda, The Netherlands; arnomacken@gmail.com (A.A.M.); denise@eygendaal.nl (D.E.); CvanBergen@amphia.nl (C.J.A.v.B.); 2Department of Orthopaedics and Sports Medicine, Erasmus University Medical Center Rotterdam, 3015 GD Rotterdam, The Netherlands

**Keywords:** clavicle, fracture, pseudarthrosis, pediatric, children, treatment, diagnosis

## Abstract

Clavicle fractures are commonly seen in the pediatric and adolescent populations. In contrast, congenital pseudarthrosis of the clavicle is rare. Although both conditions may present with similar signs and symptoms, especially in the very young, clear differences exist. Clavicle fractures are often caused by trauma and are tender on palpation, while pseudarthrosis often presents with a painless protuberance on the clavicle, which becomes more prominent as the child grows. Its presence may only become apparent after trauma, as it is usually asymptomatic. The diagnosis is confirmed on plain radiography, which shows typical features to distinguish both entities. Both clavicle fractures and congenital pseudarthrosis are generally treated conservatively with a high success rate. Operative treatment for a fracture can be indicated in the case of an open fracture, severely displaced fracture, floating shoulder, neurovascular complications or polytrauma. Congenital pseudarthrosis requires operative treatment if the patient experiences progressive pain, functional limitation and late-onset thoracic outlet symptoms, but most operations are performed due to esthetic complaints.

## 1. Introduction

Clavicle fractures frequently occur in the pediatric and adolescent populations [1]. Diagnosis and treatment of these fractures are generally straightforward but can be particularly challenging in select cases. Therefore, it is important to have a thorough understanding of the underlying principles. Furthermore, a pediatric clavicle fracture should be differentiated from congenital pseudarthrosis, which may have a similar presentation (especially in neonates) but may require a different treatment approach. Congenital pseudarthrosis of the clavicle is characterized by a failure in the fusion of the medial and lateral ossification centers of the clavicle [2]. This article aims to provide an overview of the diagnosis, treatment and complications of pediatric clavicle fractures and congenital pseudarthrosis based on the most recent literature.

## 2. Epidemiology

Clavicle fractures account for 10–15% of all pediatric fractures [1]. The majority of patients with a clavicle fracture are male (91.2%), and most clavicle fractures are seen between the ages of 10 and 19 years (incidence rate of 91.7 per 100,000) [1,3]. Fractures on the left side (58%) and on the non-dominant side (56%) are slightly more common [4]. Most clavicle fractures occur in the middle section of the bone, accounting for 70% to 95% of all pediatric clavicle fractures [1,5,6]. Displaced fractures of the clavicle are relatively common, ranging from 28% to 67% of all clavicle fractures in children and adolescents [1,4,6,7].

Clavicle fractures occur most frequently as a result of sports (66%), horseplay (12%), riding a bike (6%), a fall (6%) or another type of accident (3%) [4]. However, clavicle fractures may also occur during childbirth, particularly in the case of shoulder dystocia [8,9]. Although less than 4% of all children are born with this fracture, it is the most common fracture during childbirth, accounting for almost a third of all birth traumas [8,9,10]. 

On the other hand, congenital pseudarthrosis of the clavicle is a rare condition, and currently, available evidence relies on case reports (approximately 200 in total), with no studies reporting the incidence [2]. Congenital pseudarthrosis occurs more frequently in females and most commonly on the right side [2,6]. Isolated left clavicle pseudarthrosis occurs in less than 10%, and in most cases, presents in combination with dextrocardia or situs inversus [2]. Bilateral pseudarthrosis has been reported in about 10% of cases, often in combination with a high subclavian artery and cervical ribs or vertical upper ribs [2]. 

Congenital pseudarthrosis is often associated with abnormalities of ossification during the embryonic stage and is associated with genetic syndromes like Ehlers-Danlos, Al-Awadi/Ras-Rothschild, Kabuki and Prader-Willi [2]. 

### 2.1. Anatomy

#### Development of Clavicle

The clavicle develops from two ossification centers that are initially connected by pre-cartilage surrounded by perichondrium [2]. Physiological ossification of the clavicle occurs during the fourth week of gestation, and the two ossification centers fuse near the seventh week [2]. The epiphysis of the medial part of the clavicle does not ossify until the age of 20, and the lateral epiphysis does not ossify until the age of 25 years [1].

### 2.2. Trauma Mechanism

Most fractures are caused by blunt trauma to the shoulder or upper arm (60%), trauma to the clavicle or chest (24%) or a fall on an outstretched arm (11%) [4].

Concomitant fractures are rare in children and occur mostly in high-energy accidents involving sports or motorized vehicles [1,11]. The most common concomitant fractures are those of the ribs, spine, extremities and facial bones [1]. However, other concomitant injuries such as brachial plexopathy, compression of the subclavian vein and other neurovascular injuries are more common [1,7]. 

Another important trauma mechanism of clavicle fracture is peri-natal injury. Birth fractures are associated with shoulder dystocia and difficult delivery [8]. Risk factors for clavicle fractures are similar to risk factors related to a difficult delivery and shoulder dystocia, namely: instrumented delivery, macrosomia, post-term delivery, procedural induction of labor, prolonged labor, advanced maternal age, multiparity and excessive weight gain during the pregnancy [8]. Peri-natal clavicle fractures are often seen in combination with a fractured humerus, brachial plexus injury and injuries to the phrenic and recurrent laryngeal nerves [8]. In rare cases, an iatrogenic clavicle fracture is unavoidable to ensure successful delivery. 

### 2.3. Classification of Fractures

The Allman classification divides clavicle fractures into three groups: type 1 fractures are located in the middle third of the clavicle, type 2 fractures are located in the part lateral to the coracoclavicular ligament, and type 3 fractures are located in the medial third (Figure 1) [11,12].

### 2.4. Development of Pseudarthrosis

Pseudarthrosis of the clavicle is characterized by the incomplete or absent union of the two ossification centers [2]. Although the exact cause of pseudarthrosis is unknown, several theories have been developed as to why the fusion of the two ossification centers fails [2]. One theory is that the excessive pressure from the pulsing subclavian artery during the development of the clavicle causes non-union of the ossification centers, especially if cervical ribs are present, which add to the increased pressure [2]. Another theory is that the non-union is caused by an altered intrauterine position of the fetus and cranial localization of the right subclavian artery [2]. Additionally, rare case reports [14,15,16,17,18] of family members with pseudarthrosis suggest inheritance to attribute to the development of pseudarthrosis, although there is a lack of conclusive evidence to support this hypothesis. 

### 2.5. Classification of Pseudarthrosis

Kite proposed a classification system for congenital pseudarthrosis of the clavicle based on the differences in anatomy, clinical representation and pathology [2,19]. 

Type I includes patients who have clavicular non-union at birth, caused by hypoplasia of the distal fragment [2,19]. Pressure on the protuberance is painful, and radiographs show a larger medial fragment than lateral fragment with clear spacing between them [2,19]. For this type, the distance between the fragments and their positioning should be assessed before surgery is considered [2,19].

Type II includes patients with congenital bone deficiency who have a physiologically formed clavicle at birth which is more fragile and prone to fractures [2,19]. For this type, surgery could be considered after a fracture has occurred [2,19].

## 3. Diagnosis 

### 3.1. Clavicle Fracture

Clavicle fractures are often the result of trauma and can present as a deformity or open fracture, although visible deformity may also be absent [4,20,21]. The fracture is tender on palpation, and movement of the shoulder is labored, painful and sometimes limited [2,20]. Plain radiographs usually confirm the clinical suspicion of a fracture, yet a recent study found that it is not necessary for proper diagnosis and treatment [22]. Several studies have demonstrated ultrasound to be reliable for diagnosis of clavicle fractures both in neonates and older children [23,24,25]. One study of 58 patients found a sensitivity of 89.7% and specificity of 89.5% [25]. Ultrasound has the advantage of reducing radiation exposure but is dependent on the experience of the operator.

### 3.2. Pseudarthrosis

Pseudarthrosis of the clavicle often presents as a painless protuberance on the clavicle, most commonly in the middle third or lateral third of the bone [2]. In addition, during the first days after birth, a hypermobile segment can be seen [2]. The protuberance usually becomes larger and more evident as the child grows (Figure 2), sometimes causing atrophy of the overlying skin [2]. Furthermore, pseudarthrosis of the clavicle is often associated with a change in the alignment of the shoulder and a winged scapula [26,27,28]. This can cause a limited range of motion in all three planes, but especially when lifting the arm above the head [26,27,28]. Apart from appearance, pseudarthrosis is usually asymptomatic. However, some patients do experience pain, discomfort or functional limitations, such as late-onset thoracic outlet syndrome [2]. Although it is logical to expect the altered biomechanics to lead to a long-term impairment of the shoulder, we found no studies reporting long-term outcomes. This may be due to the low incidence of pseudo-arthrosis of the clavicle.

To confirm the diagnosis, plain radiographs need to show a clear separation between two fragments of the clavicle [2]. The fragments often have a characteristic shape towards the end facing the defect. Generally, one of the fragments appears as an “elephant’s foot” shape (the fragment is wider at the end compared to the shaft) and the other shows a “pencil point” shape (the fragment is increasingly thin towards the end) (Figure 2) [2]. The medullary canal is closed and sclerotic, but no bone callus is formed [2]. The medial fragment is often positioned superior to the lateral fragment due to muscle forces and the weight of the arm [2].

### 3.3. Differential Diagnosis

A congenital pseudarthrosis should be differentiated from a clavicle fracture. The latter is tender on palpation and is associated with a trauma or traumatic birth [2]. Old clavicle fractures can present with callus formation, which can help distinguish the difference between old and new fractures [2]. In general, pseudarthrosis is a painless protuberance (Figure 3) on the clavicle without callus formation [2]. Furthermore, several other diagnoses can have a similar presentation and should be considered in the differential diagnosis. This includes cleidocranial dysplasia, which is characterized by the absence or hypoplasia of the clavicle (usually bilateral) and presents with an increased anterior position of the shoulder but is otherwise asymptomatic [2,29]. In addition, cleidocranial dysplasia is associated with overall increased range of motion of the joints and several specific facial features (late ossification of the fontanelle, wide and protruding forehead and excess teeth) [29]. Another is neurofibromatosis, which can also cause dysplasia of the clavicle and may appear similar to a fracture or pseudarthrosis. Most of these patients have hyperpigmented “coffee stains” on their skin, pathognomonic for neurofibromatosis [2,30].

## 4. Treatment and Complications

### 4.1. Clavicle Fracture

#### 4.1.1. Non-Operative Treatment

Non-operative treatment is indicated for all fractures without displacement or other complicating factors [1,21]. The majority of clavicle fractures are treated conservatively (Figure 4b), even with significant shortening and total displacement, because children have the ability to reconstitute fracture shortening and displacement that would need surgery in adults [6,21,31,32,33,34,35,36]. To immobilize the fracture, a supportive sling, collar ‘n’ cuff or figure-of-eight bandage is prescribed for several weeks [6,21,31]. The exact length of immobilization is dependent on the severity of the fracture, the age of the child and the amount of pain [6,21,31]. The children are also instructed to avoid high-risk activities [6].

Outcomes of non-operative treatment are generally satisfactory in children and adolescents [37,38,39]. Most patients prefer the cosmetic outcome of conservative treatment [37]. However, in adolescents, conservative treatment may lead to longer functional recovery and longer time until a stable union is achieved, compared to younger children [40,41]. Non-union and mal-union are rare in children but occur slightly more frequently in the non-operative group [6,40,41].

#### 4.1.2. Operative Treatment

A small percentage of fractures require primary surgical fixation (1.6%) [6]. Fixation is indicated in the case of an open fracture, imminent open fracture, neurovascular injury, symptomatic non-union, symptomatic malunion, floating shoulder or polytrauma [1,6,34,35,42]. Relative indications for operative treatment are significantly displaced fractures (>100% of shaft width) (Figure 4a, Figure 5a and Figure 6a), severe comminution and significantly shortened fractures (> 15–20 mm absolute or > 14% relative shortening) [1,6,34,35,42,43,44,45,46,47,48,49].

The indication for surgery for fractures with significant shortening is actively discussed in the literature. Some studies have shown beneficial effects of surgery in children with a significantly shortened clavicle fracture, such as a lower incidence of mal-union and non-union [27,34,42,43,45,46,47,48,49,50,51]. However, other studies found no significant difference in outcome compared to the conservative treatment for shortened fractures [35,36,48,52]. This ambiguity is partially caused by the different methods of measuring clavicle shortening: end-to-end, cortex-to-corresponding cortex and relative shortening compared to the uninjured side [4]. Different methods may result in different cut-off values for the amount of shortening [4]. Therefore, an exact cut-off value for the amount of shortening that would be an indication for surgery cannot be concluded from the literature. In children and adolescents, clavicle shortening should be expressed in percentage shortening relative to the uninjured clavicle [4,34]. Until further consensus is reached, the choice of treatment for fracture shortening should be based on additional complicating factors, age, years of growth remaining, potential for remodeling and level of functional demand [34,42,51].

Several internal fixation methods can be used, such as plate and screw fixation (Figure 5b), screw-only fixation and intramedullary fixation (Figure 6b) [6]. Plate and screw fixation is the most commonly used technique [6]. Plate fixation has advantages over the other techniques: it provides strong fixation and compression of the small fractured fragments [44]. However, it requires an open exposure with corresponding soft tissue damage and risk of infection [44]. To reduce the size of the incision, other techniques such as the minimally invasive plate osteosynthesis (MIPO) technique, screw fixation only or intramedullary nail fixation can be used [44].

Outcomes after surgical treatment are generally satisfactory, yet not (significantly) superior to non-operative treatment [37,38,39]. There is an incongruence in the literature regarding the superiority of surgical treatment or non-operative treatment in children and adolescents. Some studies report superior outcomes in adolescents after surgery compared to non-operative treatment [40,41]. However, other studies report no clear difference in outcomes between operative and non-operative treatment in children or adolescents [6,38,39,44]. Possible advantages of surgery for adolescents are shorter recovery time, fewer cases of mal-union and non-union and shorter time to achieve union [40,41,42,43,44,47,49,51]. However, conservative treatment comes with a lower risk of complications and remains the preferred treatment in the far majority of pediatric patients.

#### 4.1.3. Revision Surgery

Revision surgery is required in the case of a refracture and non-union due to failed osteosynthesis [1,5,44]. Non-union is rare and occurs almost exclusively in patients with complete fracture displacements and refractures [5,6]. The incidence of non-union increases with increased age [6]. This may be related to skeletal maturity and more forceful trauma, which increases the chance of completely displaced fractures and concomitant injuries [6].

Bone-grafting is often used in the case of non-union, but is increasingly difficult with increased displacement [53]. Kubiak and Slongo reported that in a study of 15 patients that underwent wire or nail fixation, all patients had to undergo revision surgery [54]. Furthermore, Luo et al. reported that out of 23 patients who were surgically treated (19 with a plate and 4 with an intramedullary nail), 5 (21.7%) experienced complications (refracture, prominence of the implant and non-union due to implant failure), of whom 4 needed a revision surgery [5]. Additionally, many patients prefer to have the hardware removed due to discomfort or esthetic complaints [5].

#### 4.1.4. Return to Sports

Before returning to sports, the child should have a full range of motion, normal shoulder strength, bony healing and no tenderness on palpation [43,55]. Operative treatment could allow athletes to return to sports faster than a conservative treatment, especially for significantly displaced or shortened fractures [56,57,58]. On the contrary, in some cases, the hardware (i.e., plate, screws, pin) is removed before returning to sports, which can cause a delay [43].

On average, the time to return to sports is similar for operative and non-operative treatment because it depends on individual characteristics such as age, type and severity of the fracture and the nature of their sport [43,55,56,58].

Patients can return to non-contact sports six weeks after injury in most cases [43]. Athletes can resume contact and collision sports when solid bony union occurs, which is usually after 2–4 months [43].

### 4.2. Congenital Pseudarthrosis

#### 4.2.1. Non-Operative Treatment

The majority of patients are treated conservatively (i.e., observation only, no interventions), especially if they experience minimal symptoms and do not have esthetic complaints due to the protuberance [2,7].

Outcomes after non-operative treatment are generally excellent; most patients do not experience any pain, discomfort or limited range of motion [59].

#### 4.2.2. Operative Treatment

Indications for surgical treatment are progressive pain, functional limitation and late-onset thoracic outlet syndrome [2]. However, most operations are performed for cosmetic reasons [2]. It is recommended to perform surgery between the ages of 2 to 6 years [14,60,61].

Surgery is considered in Kite type I patients, where the fragments are less than 1 cm apart [2,19]. A displacement greater than 1 cm has a much higher incidence of non-consolidation and complications after surgery [2,19].

Several surgical treatment options are used: resection of the focus of the pseudarthrosis with the option of using a bone graft, osteosynthesis or both [2]. For stabilization, different techniques are used: an intramedullary Kirschner-wire, plate and screws, screws only, a Steinmann intramedullary pin or external fixation [2,19]. Above the age of 8 years, a bone graft is needed to achieve full consolidation [2]. The most commonly used donor site for bone grafting is the iliac crest, but the tibia, ribs and vascularized fibular grafts can also be used [2].

Post-operative treatment includes immobilization with a Velpeau sling or Desault bandage for four to six weeks [2,62].

Outcomes after surgical treatment are generally successful but appear most successful in cases with minimum or no fragment displacement and an intact periosteum and with the use of a bone graft [2].

Complications are very rare but do occur. Scar tissue may become hypertrophic, painful or form a keloid [2]. Furthermore, one case of a clavicle fracture through one of the screw holes (after the removal of the plate and screws) and one case of neuropraxia of the brachial plexus have been reported [63,64]. The most common complication is non-union, which is often an indication for revision surgery [63].

## 5. Conclusions

Clavicle fractures and congenital pseudarthrosis can be difficult to differentiate on first inspection, specifically immediately after birth. Even though pseudarthrosis of the clavicle is rare, with only a few hundred cases reported in the literature [65], it should be included in the differential diagnosis of a neonatal clavicle fracture, as undetected pseudarthrosis can cause problems at a later age. However, there are several diagnostic differences between clavicle fractures and congenital pseudarthrosis of the clavicle that can help distinguish them. Clavicle fractures are often a result of trauma, are suddenly tender on palpation and cause labored, painful or limited movement of the shoulder. Cases of congenital pseudarthrosis of the clavicle often present with a painless protuberance on the clavicle, which can become larger over time. In most cases, it is asymptomatic.

Both clavicle fractures and pseudarthrosis can be treated operatively or non-operatively, both with great success rate and patient satisfaction. Most patients with either are treated conservatively. Possible surgical indications for a clavicle fracture include an open fracture, significantly displaced fracture, shortened fracture or complications caused by the fracture. The majority of operations are successful and lasting. However, in some cases, revision surgery is required for non-union. For congenital pseudarthrosis, surgical treatment is considered in cases of progressive pain, functional limitation and late-onset thoracic outlet syndrome. However, most surgeries for congenital pseudarthrosis of the clavicle are performed because of cosmetic reasons.

In conclusion, this article provides a comprehensive, evidence-based overview of pediatric clavicle fractures and congenital pseudarthrosis. Some important issues remain open for discussion, including clear indications for surgical treatment. Most of the current knowledge is based on case studies, underpowered studies or adult-based studies. Therefore, future high-level studies in the pediatric population will need to contribute to our knowledge on these challenging pathologies.

## Figures and Tables

**Figure 1 children-09-00049-f001:**
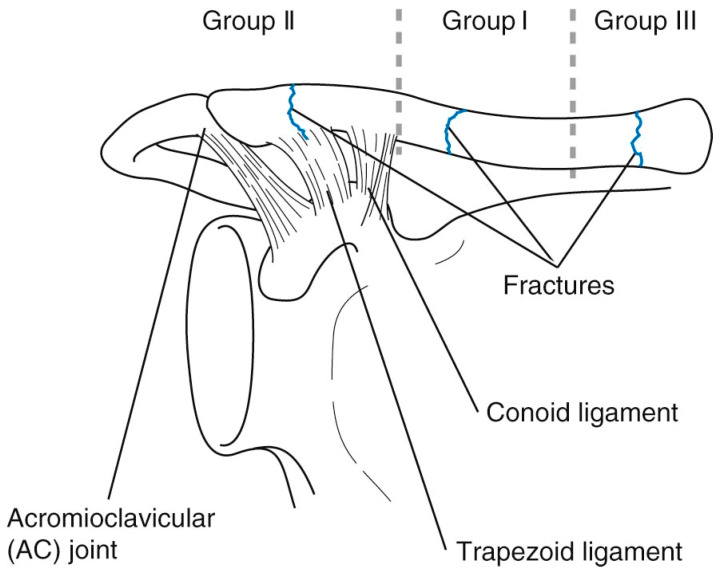
The Allman classification for clavicle fractures [13].

**Figure 2 children-09-00049-f002:**
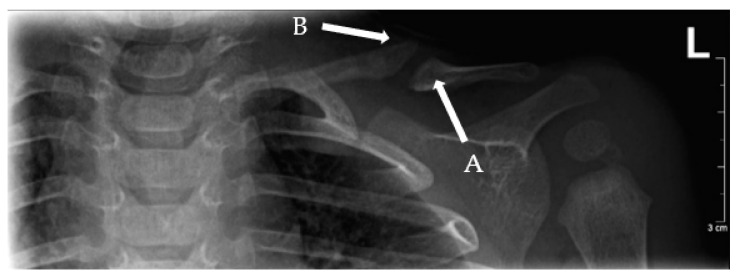
Left-sided pseudo-arthrosis (Type I) of the clavicle showing an elephant’s foot (A) and pencil point sign (B).

**Figure 3 children-09-00049-f003:**
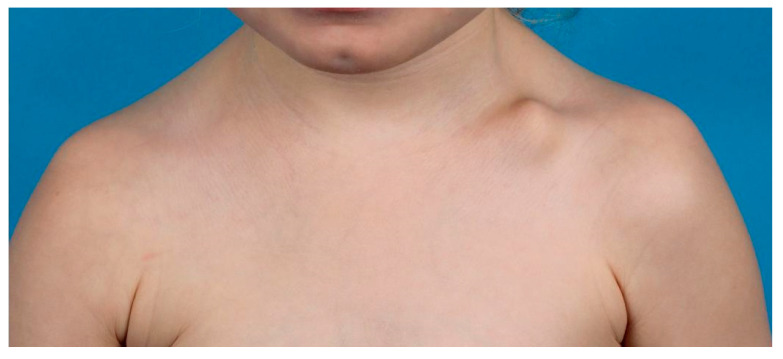
Congenital pseudo-arthrosis patient with an imminent protuberance on the left clavicle.

**Figure 4 children-09-00049-f004:**
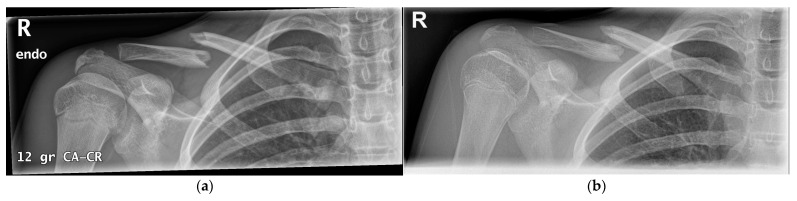
(**a**) Right clavicle fracture (Group I) with extreme displacement. (**b**) After 5 weeks of conservative treatment, early callus formation is visible.

**Figure 5 children-09-00049-f005:**
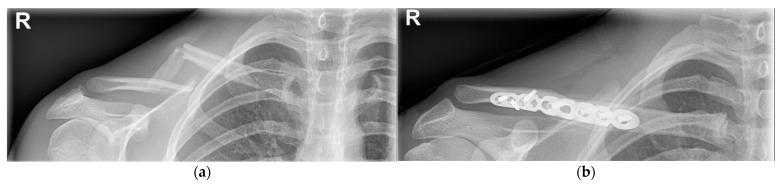
(**a**) Segmental right clavicle fracture (Group I), with extreme displacement. (**b**) Surgical fixation using the plate-and-screw method.

**Figure 6 children-09-00049-f006:**
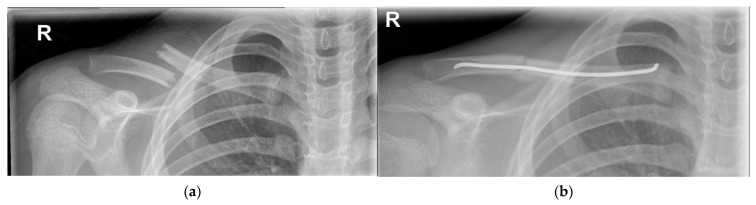
(**a**) Right clavicle fracture (Group I) with extreme displacement. (**b**) Surgical fixation with an intramedullary wire.

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
