# Peer review of "Pediatric Clavicle Fractures and Congenital Pseudarthrosis Unraveled"

_children, 2022, doi:10.3390/children9010049_

Round 1

Reviewer 1 Report

Some minor changes are needed to be considered suitable for publication.

Additional English editing is needed. The Non-Native Speakers of English Editing Certificate was not signed.

Reviewer 2 Report

This article reviewed the pediatric clavicle fractures and congenital pseudo-arthosis. Overall, it is well reviewed article. I have some suggestion as below

Line 100 classification of pseudo-arthosis

I suggest add a schematic figure to illustrate the classification as figure 1 B.

Line 117-118

It is interesting that radiography is not necessary for diagnosis. Please consider add the role of ultrasound in diagnosing clavicle fracture and associated AC joint disorder, since ultrasound had some advantages in preventing radiation exposure.

(May refer to this article, DOI: 10.1097/PEC.0b013e318235e965)

Line 126-127

From the biomechanics view, the shortened clavicle will affect the scapular movement. Is there any information regarding the long-term effect of shoulder movement in patient with pseudo-arthosis?  Will it contribute to the musculoskeletal diseases afterward?  

Line 129

The term “thoracic outlet syndrome” and “scalene syndrome” may be confusing. Scalene syndrome can also be a part of thoracic outlet syndrome due to compression to the brachial plexus by scalene muscles.

Line 130  Figure 2

Here you indicate elephant’s foot and pencil point sign, I suggest that using arrow, arrowhead or other indicator to point out the signs you mentioned.

Line 159  

The subtitle in “Treatment and complications” may confuse the readers. You may consider rearrange, including italics or bold in the whole paragraph.

Line 168-169

The order of the figure Figure 4 a.b., figure 5 a.b. and figurer 6 a.b. is not adequate. The author should put “a and b” of the same figure together, not apart it.

Line 171-173 and Line 206-208

“Articles regarding the outcomes of non-operative treatment of clavicle fractures in children is very limited, therefore we have included articles regarding adolescents to support the data from the pediatric articles.”

I suggest this illustration should be not be duplicated. You may indicate that the result was from adolescents directly in the description, and it is not necessary to clarify in the front of the paragraph.

Line 221-222

Could you explain the reason why the non-union increases with increased age? And its prevalence.

Line 266-288 

Since this is the review article, I suggest there is no necessity to have a “discussion” part. The information in these paragraphs are written in the previous section. You may directly go into the conclusion and address the important information from your review.

Reviewer 3 Report

The article describes the characteristics and treatment options for pediatric and adolescent fractures of the clavicle and pseudarthrosis of the clavicle. The articles provides a nice overview on this topic. However, a meta-analysis of the available literature on this topic would be preferable.

I recommend to change the title from “Pediatric clavicle fractures and congenital pseudo-arthrosis unraveled” to “Pediatric and adolescent clavicle fractures and congenital pseudarthrosis unraveled“.

I recommend you correct some phrases not backed adequately by the literature:

Line 15: You recommend operative treatment for fractures with shortening of the clavicle. What is the evidence for this statement in children? You know, this topic is discussed controversially, and the vast majority of displaced fractures of the clavicle in children worldwide is managed by conservative approach, with excellent results.

Line 16: How many fractures of the clavicle in polytraumatized children were stabilized in your institution within the last 5 years?

Line 226: Please avoid the word “subjects” for “child(ren)” or “patient(s)”.

Line 36: replace “10 and 19” by “10 and 19 years”

Line 78: replace “plexopathy” by “plexus injury”

Line 97: replace “claim” by “hypothesis”

Line 118: Please comment on the feasibility of ultrasound imaging to rule out fractures of the clavicle.

Line 162-167: rephrase this chapter regarding non-operative treatment. Give a differentiated description of indications for non-operative and operative treatment according to age of child, and displacement of fragments. Comment on adolescents and adolescent athletes in a separate chapter.

Please mention that many authors advocate the use of Gilchrist bandage in children.

Describe the influence of choice of treatment on the time interval between injury and “return to sports”

Provide age dependent time intervals for wearing a sling or bandage, and for return to sports. A table would be fine.

Line 162-163: Delete the statement: “without displacement or other complications”. Displacement represents no complication.

Line 182: “displaced fractures (more than one shaft-width)”. This is discussed controversially, please provide a literature search.

Line 185: replace “extreme” by “significant”

Line 214:  Provide a literature search for “Possible advantages”

Line 237: replace “positive” by “mostly excellent”

Line 245: provide incidence data if possible

Please adhere to instructions for authors when revising the “References” section. Some citations lack numbers of volume and pages (ref. 1, 4, 6, 11, 18, 19, 20, 25, 26). Provide names of editor, year of publication, pages for ref. 42.

Round 2

Reviewer 3 Report

Dear authors,

Thank you for revison, you improved the manuscript.

Johannes Mayr